# Detection of Association Features Based on Gene Eigenvalues and MRI Imaging Using Genetic Weighted Random Forest

**DOI:** 10.3390/genes13122344

**Published:** 2022-12-12

**Authors:** Zhixi Hu, Xuanyan Wang, Li Meng, Wenjie Liu, Feng Wu, Xianglian Meng

**Affiliations:** 1School of Computer Information and Engineering, Changzhou Institute of Technology, Changzhou 213032, China; 2School of Physics, Engineering and Computer Science, University of Hertfordshire, Hatfield AL10 9AB, UK; 3School of Electrical & Information Engineering, Changzhou Institute of Technology, Changzhou 213032, China

**Keywords:** Alzheimer’s disease, eigenvalues, genetic weighted random forest, fusion, significant biomarkers

## Abstract

In the studies of Alzheimer’s disease (AD), jointly analyzing imaging data and genetic data provides an effective method to explore the potential biomarkers of AD. AD can be separated into healthy controls (HC), early mild cognitive impairment (EMCI), late mild cognitive impairment (LMCI) and AD. In the meantime, identifying the important biomarkers of AD progression, and analyzing these biomarkers in AD provide valuable insights into understanding the mechanism of AD. In this paper, we present a novel data fusion method and a genetic weighted random forest method to mine important features. Specifically, we amplify the difference among AD, LMCI, EMCI and HC by introducing eigenvalues calculated from the gene *p*-value matrix for feature fusion. Furthermore, we construct the genetic weighted random forest using the resulting fused features. Genetic evolution is used to increase the diversity among decision trees and the decision trees generated are weighted by weights. After training, the genetic weighted random forest is analyzed further to detect the significant fused features. The validation experiments highlight the performance and generalization of our proposed model. We analyze the biological significance of the results and identify some significant genes (*CSMD1*, *CDH13*, *PTPRD*, *MACROD2* and *WWOX*). Furthermore, the calcium signaling pathway, arrhythmogenic right ventricular cardiomyopathy and the glutamatergic synapse pathway were identified. The investigational findings demonstrate that our proposed model presents an accurate and efficient approach to identifying significant biomarkers in AD.

## 1. Introduction

As the most common type of dementia, Alzheimer’s disease (AD) affects the cognition, behavior, and memory of senior adults [1]. In the research of AD, biomarkers can provide new targets for AD and facilitate the diagnosis and treatment of AD. Recently, scientists have obtained a large amount of imaging data (such as the Magnetic Resonance Imaging (MRI), Computed Tomography (CT) and Diffusion Tensor Imaging (DTI)) and the corresponding genetic data (such as the genetic sequencing data). These imaging data and genetic data were used to explore the association between them, detect the biomarker and diagnose the early stage of the disease. Genome-wide association analysis (GWAS) was proposed to identify single nucleotide polymorphisms (SNPs) associated with phenotype [2]. On this basis, FGWAS [3], vGWAS [4], vGeneWAS [5] and other methods [6] were developed to identify genetic markers. To avoid the disadvantages of the single GWAS method, studies based on GWAS and machine learning were performed to capture the markers. Compared with pure association analysis, the combination of GWAS method and machine learning was more effective in analyzing the association between imaging data and genetic data.

Gaetani et al. [7] identified that SIRT2, HGF, MMP-10 and CXCL5 reflected neuroinflammation in early AD using cerebrospinal fluid analysis, spearman correlation and least absolute shrinkage and selection operator. Popuri et al. [8] proposed a model to predict the probability of individuals diagnosed with dementia of Alzheimer’s type (DAT) and achieved the classification area under the curve (AUC) of 0.78 from individuals diagnosed with DAT and from individuals not diagnosed with DAT. Using multimodal neuroimaging biomarkers, Luckett et al. [9] applied artificial neural networks to investigate the changes in Aβ deposition, glucose metabolism and brain atrophy of 131 mutation carriers and 74 non-carriers. They obtained the caudate, cingulate, and precuneus as predictors and identified the biphasic response in metabolism. Sinead et al. [10] used 304 subjects from the INSIGHT-preAD cohort to construct the candidate features and applied the random forest method, logistic regression, and support vector machine to predict the amyloid status. Ezzati et al. [11] used the K-nearest neighbors (KNN) algorithm to identify individuals with declining cognitive function and stable cognitive function among the 202 participants from the Alzheimer’s Disease Neuroimaging Initiative (ADNI) and the 77 participants from the placebo arm of the phase III trial of Semagacestat. They obtained a positive prediction accuracy of 80.8%. However, there are still limitations in these studies. Although machine learning methods have been applied and promising results found, the machine learning models used in these studies were traditional models, and there is no performance comparison against the other methods. For example, if we want to extract an important feature set from thousands of features, hundreds of thousands of experiments should be repeated using traditional models and this will lead to a large cost of time. The fusion model of different algorithms provides an efficient method to resolve the problem. Moreover, due to the large number of features generated from voxel-based MRI, how to refine and fuse the multi-features and extract effective information is a field worthy of further investigation.

The workflow of this study is shown in Figure 1.

In this study, we have proposed a novel method to fuse multi-features and a genetic weighted random forest model to identify the important fused features (calculated by the eigenvalues from genetic data and the voxel values from voxel-based MRI data) from different datasets. Firstly, we constructed a matrix using the genetic data associated with AD and calculated the eigenvalues of the matrix using the datasets of HC and AD. Then, we extracted the voxel features from imaging data and applied the eigenvalues of genetic data to construct the fused features. Subsequently, we proposed a genetic weighted random forest model to mine the important features from the fused features and applied different models for modeling comparison. We assessed the biological significance using the extracted features. Finally, we introduced the EMCI and LMCI datasets to evaluate the changes of genes and pathways that might lead to the transaction from EMCI to AD. The results prove that our model is efficient to extract the important features and to provide the candidate biomarkers for the AD diagnosis. In addition, our method can be used for other neurological diseases.

## 2. Materials and Methods

### 2.1. Imaging and Genetic Data

In this study, we used the imaging and genetic data for the prediction of AD biomarkers. We downloaded the MRI data of 680 males and 587 females, including 310 HC, 271 EMCI, 390 LMCI and 296 AD subjects from ADNI (adni.loni.usc.edu, accessed on 15 May 2020). The details of these participants are shown in Table 1.

We applied voxel-based morphometry (VBM) to preprocess the MRI scans. The downloaded images were skull-stripped and segmented using CAT12 [12]. Then, panning, rotating and swivel were applied to the resulting images [13]. The images were then subjected to local nonlinear deformation and aligned to the Montreal Neurological Institute (MNI) space. To ensure uniformity of the imaging data for each participant and to preserve inter-individual variation, we modulated the images and obtained the gray matter images. Finally, to reduce the cost of data processing, we down-sampled the obtained images from 181 × 218 × 181 to 61 × 73 × 61 voxels and aligned them into the anatomical automatic labeling (AAL) template [14,15].

The Illumina GWAS arrays (610-Quad, OmniExpress or HumanOmni2.5-4v1) (Illumina, Inc., San Diego, CA, USA) and blood genomic DNA samples were used to genotype the participants [16]. Then, we applied PLINK v1.9 [17] to extract SNPs using the following process: (1) extracting SNPs on chromosome 1–22; (2) call rate of each SNP ≥ 95%; (3) minor allele frequency of each SNP ≥ 5%; (4) Hardy–Weinberg equilibrium test *p* ≥ 1 × 10^6^; (5) call rate of each participant ≥ 95%. Finally, we obtained 5,574,300 SNPs that passed the quality control.

In this study, we used the principal components from population stratification analysis as the covariates for GWAS (linear regression). We separated the participants into four groups based on the diagnosis statues of the participants. To determine the optimal number of principal component analysis (PCA) in our study, we conducted GWAS repeatedly with different numbers of PCA ranging from 1 to 20. Then, we applied the GATEs (gene-based association test that uses extended Simes) method [18] to calculate *p*-values for genes. Due to the biological significance of APP, we selected APP as the candidate gene to determine the optimal number of PCA. Figure 2 shows the results of APP with different numbers of PCA.

From Figure 2, we can observe that the best number of PCA in HC is 16, while the number in EMCI, LMCI and AD are 17, 13 and 17, respectively. With the optimal parameters, we performed GWAS (linear regression) using the genotype, phenotype, and covariates (including age, gender, education, and the optimal number of PCA) in each group and conducted Bonferroni correction for multiple testing. Then, we obtained the GWAS results with 5,574,300 SNPs. We selected SNPs with a *p*-value less than 0.05 and mapped them into corresponding genes.

### 2.2. Features Fusion

To construct the fused features, we analyzed the MRI and genetic data for each group separately. For the MRI data, we used the 90 brain regions from AAL atlas to calculate the voxels of each participant and saved them as matrices ***M***. The voxel-based MRI data of 1267 participants was used in our work and the number of voxels in 90 brain regions was 43,851. Since there were a total of 43,851 voxels in 90 brain regions, the size of ***M*** was 1267 × 43,851. For our binary classification, two sets of data were required in our experiment. So, we obtained the $M_{AD}$, $M_{LMCI}$, $M_{EMCI}$ and $M_{HC}$ from ***M*** according to Table 1, while the size of $M_{AD}$, $M_{LMCI}$, $M_{EMCI}$ and $M_{HC}$ were 296 × 43,851, 390 × 43,851, 271 × 43,851 and 310 × 43,851, respectively.

For the genetic data, we calculated SNPs in 24 genes [19] associated with AD. Then, we used the corresponding SNPs to construct the vector $v_{SNP}$ (Equation (1)).
(1)$$v_{SNP}=\left\{\begin{array}{c} \begin{array}{ccc} {\mathrm{SNP}}_{1,1} & {\mathrm{SNP}}_{1,2} & \begin{array}{cc} \ldots & {\mathrm{SNP}}_{1,j} \end{array} \end{array}\\ \begin{array}{ccc} {\mathrm{SNP}}_{2,1} & {\mathrm{SNP}}_{2.2} & \begin{array}{cc} \ldots & {\mathrm{SNP}}_{2,j} \end{array} \end{array}\\ \begin{array}{c} \vdots \\ \begin{array}{ccc} {\mathrm{SNP}}_{i,1} & {\mathrm{SNP}}_{i,2} & \;\;\begin{array}{cc} \ldots & {\mathrm{SNP}}_{i,j} \end{array} \end{array} \end{array} \end{array}\right\},\;\;\;\;\;\;\;\;\;\;\;\;\;\;\;\;\;\;\;\;i,j\in \left[1,\;24\right]\;\;\;\;\;\;\;$$
where ${\mathrm{SNP}}_{i,j}$ is the *j*th SNP in $Gene_{i}$. The vector $v_{SNP}$ contained information of SNPs in 24 genes and used the *p*-value to generate vector $v_{P}$ (Equation (2)).
(2)$$v_{P}=\left\{\begin{array}{c} \begin{array}{ccc} P_{1,1} & P_{1,2} & \begin{array}{cc} \ldots & P_{1,j} \end{array} \end{array}\\ \begin{array}{ccc} P_{2,1} & P_{2.2} & \begin{array}{cc} \ldots & P_{2,j} \end{array} \end{array}\\ \begin{array}{c} \vdots \\ \begin{array}{ccc} P_{i,1} & P_{i,2} & \;\;\begin{array}{cc} \ldots & P_{i,j} \end{array} \end{array} \end{array} \end{array}\right\},\;\;\;\;\;\;\;\;\;\;\;\;\;\;\;\;\;\;\;\;\;\;\;\;i,\;j\in \left[1,\;24\right]\;\;\;\;\;\;\;\;\;\;$$

From Equation (2), we obtained the vector ($v_{P}$) in each group. Since the number of SNPs in some genes were less than 24, we filled in the missing values with 0. Using the resulting vectors, we obtained the matrix ($M_{P}$) from Equation (3).
(3)$$M_{P}=\left[\begin{array}{c} \begin{array}{ccc} v_{P_{1,1}} & v_{P_{1,2}} & \begin{array}{cc} \ldots & v_{P_{1,j}} \end{array} \end{array}\\ \begin{array}{ccc} v_{P_{2,1}} & v_{P_{2,2}} & \begin{array}{cc} \ldots & v_{P_{2,j}} \end{array} \end{array}\\ \begin{array}{c} \vdots \\ \begin{array}{ccc} v_{P_{i,1}} & v_{i,2} & \;\;\begin{array}{cc} \ldots & v_{P_{i,j}} \end{array} \end{array} \end{array} \end{array}\right],\;\;\;\;i,j\in \left[1,\;24\right]$$

Then, we applied Equation (4) to calculate the eigenvalue of each $M_{P}$.
(4)$$\left(\alpha E-M_{P}\right)x=0$$
where ***E*** is the unity matrix, *x* is an eigenvector, and α is the corresponding eigenvalue. Since there were 24 eigenvalues in each group, we selected the max(α) as the final eigenvalue of each group and defined them as $\alpha_{AD}$, $\alpha_{LMCI}$, $\alpha_{EMCI}$ and $\alpha_{HC}$.

Using the resulting max(α) and ***M***, we defined the initial datasets as ***S***, while the elements of them were calculated by α×M. Let $S_{AD-HC}$, $S_{AD-EMCI}$, $S_{AD-LMCI}$, $S_{LMCI-HC}$ and $S_{EMCI-HC}$ denote the five initial datasets. These datasets were defined as Equation (5).
(5)$$\begin{array}{c} S_{AD-HC}=\left[\left[S_{AD}\right],\;\left[S_{HC}\right]\right]\\ \begin{array}{c} S_{AD-EMCI}=\left[\left[S_{AD}\right],\;\left[S_{EMCI}\right]\right]\\ S_{AD-LMCI}=\left[\left[S_{AD}\right],\;\left[S_{LMCI}\right]\right]\\ S_{LMCI-HC}=\left[\left[S_{LMCI}\right],\;\left[S_{HC}\right]\right] \end{array}\\ S_{EMCI-HC}=\left[\left[S_{EMCI}\right],\;\left[S_{HC}\right]\right] \end{array}$$
where $S_{AD-HC}$, $S_{AD-EMCI}$, $S_{AD-LMCI}$, $S_{LMCI-HC}$ and $S_{EMCI-HC}$ were 606 × 43,851, 567 × 43,581, 686 × 43,581, 700 × 43,581 and 581 × 43,581, respectively.

### 2.3. Genetic Weighted Random Forest Construction

In this study, we applied the genetic weighted random forest method to extract the important features contributing to the classification of different groups. The workflow of the genetic weighted random forest method is shown in Figure 3.

Definition of the $S_{train}$, $S_{valid}$ and $S_{test}$.

In this model, we introduced genetic algorithm and weight to a traditional random forest. Specifically, taking the $S_{AD-HC}$ as an example, we selected features randomly from dataset $S_{AD-HC}$ to construct the initial random forest. The dataset $S_{AD-HC}$ was defined as Equation (6).
(6)$$S_{AD-HC}=\left\{feature_{i},label_{i}\right\},\;\;\;i\in \left[1,\;N\right]$$
where $feature_{i}$ is the features of sample *i* in $S_{AD-HC}$, $label_{i}$ is the label (“1” or “−1”) of the corresponding features and *N* is the sample number. The label corresponding to HC is “1”, the label corresponding to AD samples is “−1”, and *N* is 606.

To find the significant features, we separated the ***S*** into $S_{train}$, $S_{valid}$ and $S_{test}$ with the ratio defined in Equation (7).
(7)$$S_{train}:\;S_{valid}:\;S_{test}=6:2:2$$

Construction of the decision tree.

To construct the initial decision trees, we randomly selected the features from Equation (8).
(8)$$tree_{f}=feature\left(fix\left(\sqrt{num}\right)\right),\;\;\;f\in \left[1,\;n\right]$$
where $tree_{f}$ is the features in decision tree *f*, feature is the features in $S_{AD-HC}$, fix() is the upward rounding function and num is the number of features in $S_{AD-HC}$. Then, we obtained one decision tree. The step was repeated for *n* times to obtain *n* decision trees, and the initial random forest was composed of *n* decision trees.

Genetic evolution.

The initial random forest was regarded as the initial population. The decision tree in each group with the best classification accuracy in $S_{valid}$ was picked out for the genetic evolution and two groups of five decision trees were selected as the parents. A new decision tree was generated from the selected parents. Then, the genetic evolution was repeated for *n* times and *n* new decision trees were generated. By repeating the steps above for (1, 50) times, we obtained a new random forest. 

Weight calculation.

The classification accuracy of decision tree was defined as Equation (9).
(9)$$ACC_{f}=N_{vf}/N_{v},\;\;\;f\in \left[1,\;n\right]$$
where $ACC_{f}$ is the classification accuracy of decision tree *f* in $S_{valid}$, $N_{vf}$ is the number of samples that classified correctly by tree *f* in $S_{valid}$, and $N_{v}$ is the number of samples in $S_{valid}$.

Weighted decision tree.

The accuracy of each resulting decision tree was used as the weight $W_{f}$ to adjust the corresponding decision tree (Equation (10)).
(10)$$WTree_{f}=\;W_{f}\times tTree_{f},\;\;\;f\in \left[1,\;n\right]$$
where $tTree_{f}$ is the resulting decision trees, $WTree_{f}$ is the weighted decision tree *f* and $W_{f}$ is the weight of decision tree *f*. To obtain the final classification accuracy of the genetic weighted random forest, we utilized $S_{test}$ to calculate the classification accuracy using Equation (11).
(11)$$ACC_{test}=\frac{N_{tw}}{N_{t}}$$
where the $ACC_{test}$ is the final classification accuracy of the genetic weighted random forest, the $N_{tw}$ is the number of samples that classified correctly by all weighted decision trees in $S_{test}$ and the $N_{t}$ is the number of samples in $S_{test}$.

### 2.4. Comparison of Different Models

To determine the repeatability of our model, we performed 10 independent experiments. We also applied the traditional random forest, genetic evolution random forest and weighted random forest as the comparison models to verify our model. Moreover, to evaluate the stability of our model, we performed 10 independent experiments and calculated the accuracy with 5 datasets.

### 2.5. Biological Significance Assessment

Through the process described in Section 2.3, we obtained five sets of features from $S_{AD-HC}$, $S_{AD-EMCI}$, $S_{AD-LMCI}$, $S_{LMCI-HC}$ and $S_{EMCI-HC}$. To extract the important features, we sorted the features from $S_{AD-HC}$ and $S_{EMCI-HC}$ according to their frequency and selected the top 1000 features as candidate features. Then, we defined the range of features as [*k*, 1000], where *k* was the smallest integer greater than $fix\left(\sqrt{num}\right)$ and divisible by 10, and the step was 10. Subsequently, we applied the genetic weighted random forest to select the important features.

Using the resulting features, we calculated the optimal number of PCA and performed GWAS with covariates (age, gender, education and PCA results). The ECS (Effective chi-square test) method [20] was applied to calculate the *p*-values of genes and Bonferroni correction was applied to perform multiple testing. Finally, we selected the genes (corrected *p*-value < 0.001) and introduced the pathway analysis [21] to assess the biological significance of the important features.

## 3. Results

### 3.1. The Results of Parameter Optimization

We used the $S_{AD-HC}$ as the dataset ***S***. Initially, $\sqrt{43,581}\approx 209$ features were randomly extracted from the initial datasets to construct one initial decision tree. Then, the number of decision trees *f* was set to the range of (300, 500) and *f* decision trees were randomly constructed. We randomly selected and traversed all the parameters described in Section 2.3. To find the optimal parameter of each model, we conducted 12 repeated independent experiments. To avoid the accidental best parameter, we removed the best and the worst results. The accuracy and the corresponding number of decision trees are shown in Figure 4. The corresponding ROC curves (receiver operating characteristic curve) are shown in Figure 5.

As shown in Figure 4, all of the peaks of classification accuracy in four models are 80.99%. The nodes of random forest and weighted random forest are 460, while the nodes of genetic evolution random forest and genetic weighted random forest are (340, 32) and (320, 50). From Figure 5, we find that the AUC values (Area Under Curve) of four models are all above 0.8 and our model has the best AUC of 0.838. Then, we applied the optimal parameters to obtain the final classification accuracy in $S_{test}$. The 10 independent experiments were used to ensure greater credibility in our findings. Figure 6 shows the results of 4 models in $S_{test}$ and Figure 7 shows the ROC curves of the four models in $S_{test}$. Figure 8 shows the boxplot of 4 models in 10 independent experiments.

As shown in Figure 6 and Figure 8, the best accuracy is found in our model, and the accuracy of our model is above 91%. The best is 93.44% and the worst is 91.8%. The genetic evolution random forest is lower than our model while the genetic evolution is applied in them. From Figure 7, we find that the AUC values of four models are all above 0.92. Moreover, the random forest model and weighted random forest model have the same classification accuracy seven times, and the other three times, the accuracy of the weighted random forest model is better than the random forest model. This shows that although the use of weights contributes to classification, the improvement is limited. When the genetic evolution was introduced in the random forest, the accuracy changed a lot. There are six times that the accuracy worsens, one time that the accuracy is the same, and only 3 times that it gets better. This shows that genetic evolution introduces changes to decision trees, but the introduced change is not necessarily positive. However, the use of weights after genetic evolution greatly increases the accuracy. 

We also calculated the Precision, Recall and F1 score and present them in Table 2.

### 3.2. Extraction of Important Features

To evaluate the universality of our model and to avoid the contingency of the experiment, we conducted 10 independent experiments with the other 4 datasets ($S_{AD-EMCI}$, $S_{AD-LMCI}$, $S_{LMCI-HC}$ and $S_{EMCI-HC}$). Figure 9 shows the accuracy of the 10 independent experiments with all 5 datasets. Figure 10 shows the boxplot of our models in five datasets. Table 3 presents the Precision, Recall and F1 score of our model in 10 independent experiments.

As shown in Figure 9 and Figure 10, the accuracy of $S_{AD-HC}$ is 93.44% and the worst one is 91.8%. The difference between them is only 1.64%. The results with $S_{AD-EMCI}$ show better stability (the best is 93.86% and the worst is 92.98%), while the results with $S_{AD-LMCI}$ also give a difference with 1.45% (the best is 94.2% and the worst is 92.75%). The accuracy with $S_{LMCI-HC}$ is the best among the five datasets (the best is 95% and the worst is 92.85%). The best accuracy with $S_{EMCI-HC}$ is 80.34% and the worst is 76.92%, which is also the worst in five datasets. However, the differences in the best and worst accuracy results are less than 4% for all five datasets. This proves that that the results are stable, and there is no experimental contingency in the results.

From Section 3.1, we obtained the final features using our model. However, the features were too many for the biological significance assessment. We needed to extract the features with the best classification and refer to them as important features. To determine the important features, we calculated the frequency of all features obtained and sorted them according to their frequency. Then, we used the top 1000 features as the candidate features and used our model to select the important features. Considering that there were 209 nodes in each of our decision trees, we set the new number of nodes in the range of [210, 1000] and the step size to 10. Subsequently, we applied the parameters obtained in Section 3.1 to construct the initial random forest. The classification accuracy of the features is shown in Figure 11.

From Figure 11, we can observe that the best accuracy was 90.98% with 880 features. So, the top 880 features were the important features in $S_{AD-HC}$. Moreover, considering that the best accuracy of 95% is in $S_{LMC-HC}$ and the worst is in $S_{EMCI-HC}$, we also extracted the important features in $S_{LMCI-HC}$ and $S_{EMCI-HC}$ and found that the best classification accuracy of $S_{LMCI-HC}$ was 97.86% and $S_{EMCI-HC}$ was 76.07%. The corresponding node were 800 and 280, respectively. So, the important features of $S_{LMCI-HC}$ were the top 800 features and the important features of $S_{EMCI-HC}$ were the top 280.

### 3.3. Biological Significance Assessment

Using the extracted features, we calculated the optimal PCA number of each group. Figure 12 shows the *p*-value and the PCA number. 

From Figure 12, we find the optimal PCA number in $S_{AD-HC}$, $S_{LMCI-HC}$ and $S_{EMCI-HC}$ is 5, 3 and 8, respectively. Using the selected number, we performed GWAS in each group and calculated *p*-values for genes. Then, we applied the genes with corrected *p*-values < 0.001 (Bonferroni corrected) for pathway analysis. We selected the pathways with corrected *p*-value < 0.01 in each group. The pathways and their corrected *p*-value of each group are shown in Figure 13 [22] and the top 20 genes in 3 groups are shown in Figure 14.

As shown in Figure 13, there are three intersection pathways with corrected *p*-value < 0.01 in EMCI−group, LMCI−HC group and AD-HC group, while the three pathways are the significant pathways related to AD. This indicates that there are some common pathways among the EMCI, LMCI and AD groups, possibly because the EMCI and LMCI are the transaction statues between HC and AD. In addition, we also find that the number of top pathways is quite different in each group, suggesting that this may be caused by the different top genes with corrected *p*-value < 0.001 in each group. Therefore, we calculated the top 20 genes of each group and presented them in Figure 14.

From Figure 14, we can observe that there are five of the same top genes in three groups, and these genes are all associated with AD. The numbers of genes in only one group are 10 (EMCI−HC), 8 (LMCI−HC) and 9 (AD−HC), and the numbers of genes between two groups are 3 (EMCI−HC and LMCI−HC), 4 (LMCI−HC and AD−HC) and 2 (EMCI−HC and AD−HC).

## 4. Discussion

In this study, we proposed a novel feature construction method using the eigenvalues of gene matrix to amplify the features of each group. Moreover, we also proposed a feature detection model, which was used to mine the underlying information of the fused features.

From Figure 2, we find that the *p*-values changed depending on the number of principal components. However, due to the diversity of imaging data and genetic data in each group, the change was not a linear process and the GWAS result of each group was also quite different. This indicated that the difference of data in each group was amplified by jointly analyzing imaging and genetic data. Some researchers used the gene data to fuse features in other methods [23,24,25] and obtained satisfactory classification performance. In this study, we selected the GWAS results associated with AD to construct a matrix and calculated the eigenvalues to adjust the imaging features. Using the resulting features as the fused features, we proposed a genetic weighted random forest to mine the features contributed to AD.

As shown in Figure 4, we find that the optimal numbers of initial random forests of 4 models is 460, 460, 340 and 320, respectively. The training process and training results in random forest model and weighted random forest are roughly the same. The difference between the two models is used as an additional weight coefficient in the weighted random forest. In the 10 independent experiments of $S_{test}$, we also found that the accuracy of weighted random forest is better than random forest. This indicates that the weight is able to improve the performance of random forest model. When the genetic evolution was added into random forest instead of the weight, the accuracy of the model fluctuated and no trend was found. In $S_{test}$, the genetic evolution random forest did not perform as well as it did with $S_{valid}$. However, the accuracy became the best in 10 independent experiments when the weight and genetic evolution were added into the random forest. Although genetic evolution did not improve the performance of the decision trees, it changed the features of the decision trees, and the addition of weight made a qualitative change in the model. The accuracy of the new model reached 93.44% and the fluctuation of accuracy was less than 2% in the 10 independent experiments. This suggests that the combination of genetic evolution and weight greatly enhanced the performance and stability of the model. To evaluate the universality of our model, we performed another 10 independent experiments with $S_{AD-EMCI}$, $S_{AD-LMCI}$, $S_{LMCI-HC}$ and $S_{EMCI-HC}$. From Figure 9, we observe that although the application of genetic data amplifies the differences between imaging features, the accuracy of $S_{EMCI-HC}$ is the lowest. This may be caused by the smaller difference in features between the EMCI and HC groups. The accuracy of the other four datasets is all above 90% and the gaps of accuracy in the four datasets are all approximately 2%. This proves that our model outperforms the other three models and the features of AD are quite different from the other three groups, as well as the features of the LCMI and HC groups. As shown in Figure 11, we identified the important features of three datasets and the accuracy of the important features were 76.07%, 97.86% and 90.98%, which were similar to the results obtained from $S_{test}$. This proves that the results mined by our model are valid candidates for filtering important features. Likewise, there are other neurological diseases for which imaging data and genetic data are available. By fusing the corresponding voxel-based image data and genetic data, it is certainly the case that our model could also be applied effectively to these diseases and we expect to develop further deep learning-based models for a wider range of diseases.

By analyzing Figure 13 and Figure 14, the *CSMD1* was found to be related to AβPP metabolism and AD [26]. The decreasing of the *CDH13* expression level reduced the cell apoptosis in AD [27]. Taylor et al. found that the *PTPRD* altered in AD and the *PTPRD* mediated by *BACE1* provided potential important new mechanisms for AD risks [28]. In another study, *PTPRD* was identified as the significant gene associated with AD [29]. The *MACROD2*, which was the neurodevelopmental-related gene, was reported as the risk loci of autism spectrum disorder [30], while the autism spectrum disorder had overlapping mechanisms of pathogenesis with AD [31]. Therefore, we suggested that the *MACROD2* was associated with AD by affecting the autism spectrum disorder. The *WWOX* was down-regulated in the hippocampus of AD patients [32] and conferred AD risk [33]. Sze et al. reported that the reduction of *WWOX* modulated the generation of Aβ [32,34]. Furthermore, we also identified specific genes in each group. The *CAMTAs* were bind calmodulin to activate transcription [35] and the calmodulin played a role in long-term potentiation, learn and memory [28]. The *CTNNA2* was found to be related to general cognitive function by analysis of GWAS data [36]. Li et al. found that there were three mutations of *SOX5* associated with AD by segregating with the affection status, and they suggested that the *SOX5* might be a new candidate gene of AD [37].

We found that the EMCI-HC group had more pathways with corrected *p*-value < 0.01 than LMCI-HC and AD-HC groups. The pathways were constantly streamlined from EMCI-HC and LMCI-HC to AD-HC group. The reason might be that EMCI and LMCI were the precursor stages of AD, and the increasing difference in each group might be another reason for pathway reduction. We also found that there were three intersection pathways in the three groups, and the calcium signaling pathway (corrected *p*-value = 9.83 × 10^6^ in EMCI-HC, corrected *p*-value = 1.9 × 10^3^ in LMCI-HC and corrected *p*-value = 2.54 × 10^4^ in AD-HC) was the significant pathway. For example, excessive Ca^2+^ played a role in AD by increasing Aβ level, and the Ca^2+^ could be increased by Aβ, too [38,39,40,41]. The arrhythmogenic right ventricular cardiomyopathy (corrected *p*-value = 4.75 × 10^5^ in EMCI-HC, corrected *p*-value = 3.8 × 10^3^ in LMCI-HC and corrected *p*-value = 1.63 × 10^3^ in AD-HC) was mapped to the region of chromosome 14q24 using linkage analysis, while one of the early-onset AD genes (AD3) was identified in this region [42]. For the glutamatergic synapse pathway (corrected *p*-value = 1.37 × 10^4^ in EMCI-HC, corrected *p*-value = 2.95 × 10^3^ in LMCI-HC and corrected *p*-value = 4.78 × 10^3^ in AD-HC), the glutamate played an important role in memory and learning and the absence of glutamate could affect the memory, cognition, and behavior [43,44]. As the receptor of glutamate, N-methyl-D-aspartic-acid could be identified by inducting long-term potentiation (corrected *p*-value = 1.52 × 10^3^ in EMCI-HC) and long-term depression (corrected *p*-value = 4.78 × 10^3^ in AD-HC) [45,46]. The changes in synaptic strength were caused by the long-term potentiation and long-term depression and affected memory and learning [47]. The connection between memory and long-term potentiation in the hippocampus was identified by molecular genetic approaches [48] and tau phosphorylation in the hippocampus was enhanced by the induction of long-term depression [49,50]. From EMCI to AD, the long-term potentiation was not found in the AD-HC group and the long-term depression became a significant pathway. We suggested that the changes of these pathways might be the induced reason of the transaction from EMCI to AD.

## 5. Conclusions

In this paper, imaging-genetics research was performed to detect important features using genetic weighted random forest. Specifically, a novel feature fusion method was proposed to construct the fused features. Moreover, we analyzed the significance of the extracted features and proved that our model was able to produce promising performance in screening the potential features. We evaluated the robustness and stability with other data groups and this suggested that our model could be used in other diseases. By analyzing the results, we identified the intersection genes in the EMCI-HC group, LMCI-HC group and AD-HC group, including *CSMD1*, *CDH13*, *PTPRD*, *MACROD2* and *WWOX.* We also identified some significant pathways associated with AD, such as the calcium signaling pathway (corrected *p*-value = 9.83 × 10^6^ in EMCI-HC), arrhythmogenic right ventricular cardiomyopathy (corrected *p*-value = 4.75 × 10^5^ in EMCI-HC and glutamatergic synapse pathway (corrected *p*-value = 1.37 × 10^4^ in EMCI-HC). The findings demonstrate that our proposed method is able to outperform several important biomarkers for the transaction of AD diagnosis and to identify potential biomarkers for other diseases. Naturally, our work has some limitations. For example, deep learning is a new field of machine learning that uses a machine learning technique called artificial neural networks to extract features and make predictions. The application of deep learning may improve the performance of our work. We will explore a deep learning approach in our future work to investigate our proposed ideas.

## Figures and Tables

**Figure 1 genes-13-02344-f001:**
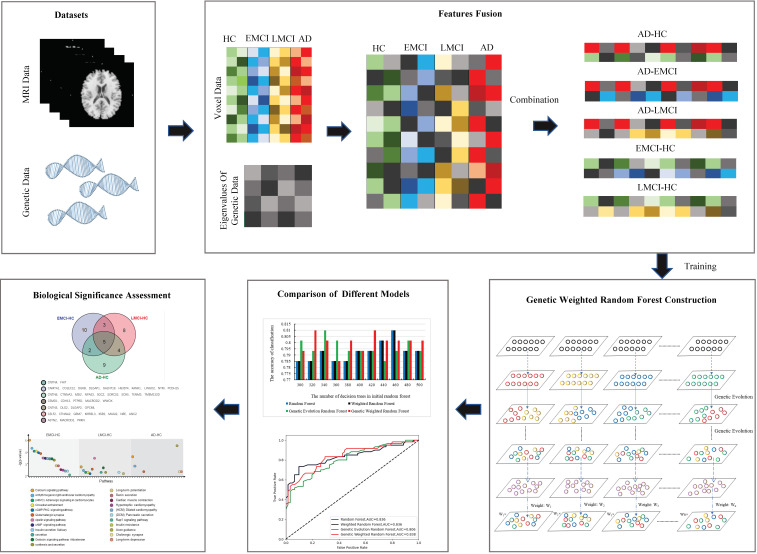
The workflow of the presented study.

**Figure 2 genes-13-02344-f002:**
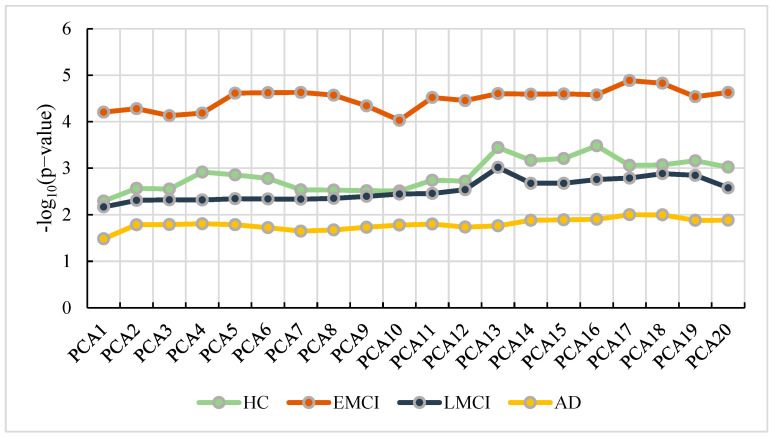
The number of PCA vs. the corresponding *p*-value of APP.

**Figure 3 genes-13-02344-f003:**
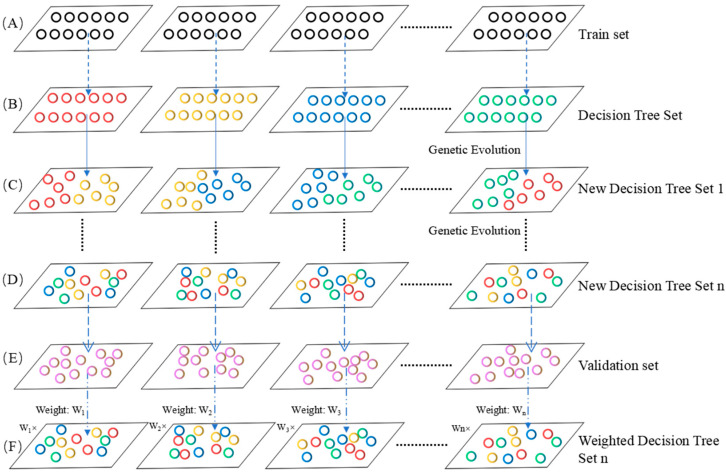
The workflow of genetic weighted random forest. (**A**) The training set. (**B**) The decision tree set constructed using the train set. (**C**) The new decision tree set after genetic evolution for once. (**D**) The new decision tree set after genetic evolution for *n* times. (**E**) The calculation of weight of each decision tree in validation set. (**F**) The weighted decision tree using the weight from E.

**Figure 4 genes-13-02344-f004:**
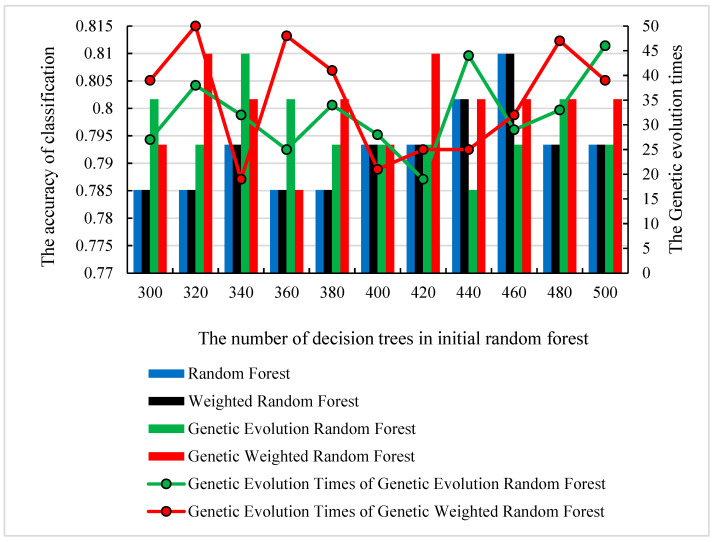
The optimal number of decision trees, genetic evolution times and their corresponding classification accuracy.

**Figure 5 genes-13-02344-f005:**
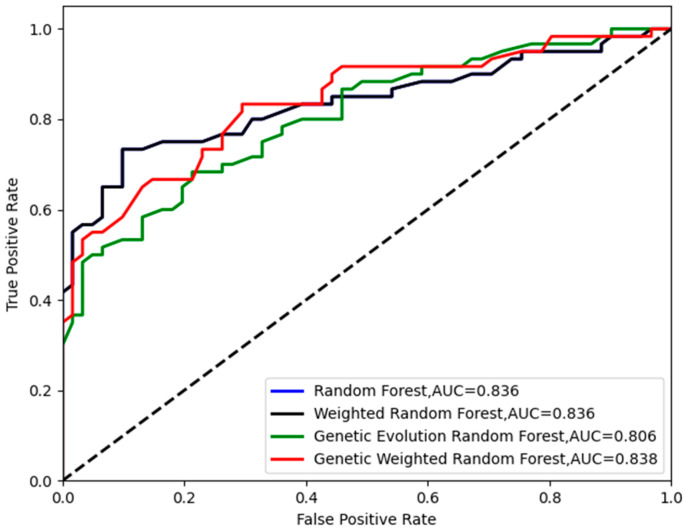
The ROC curves of the four models in $S_{valid}$.

**Figure 6 genes-13-02344-f006:**
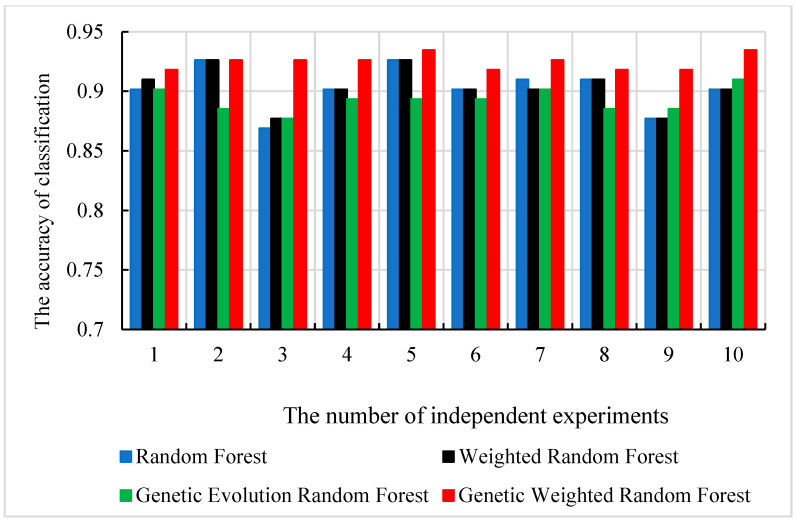
The classification accuracy of 4 models in 10 independent experiments.

**Figure 7 genes-13-02344-f007:**
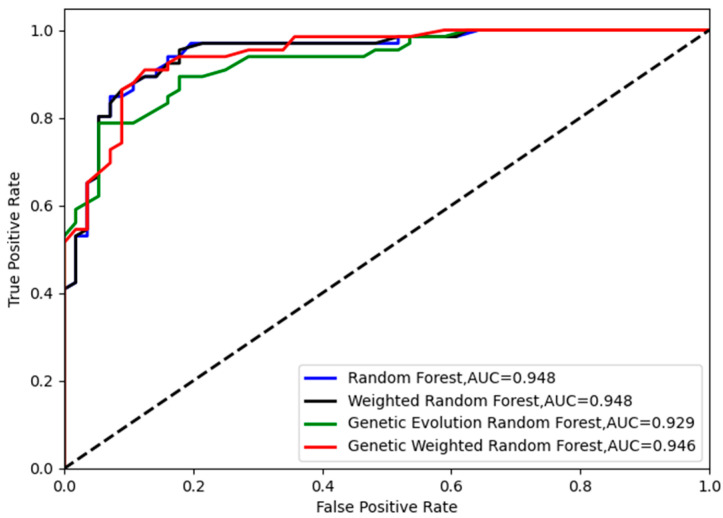
The ROC curves of the four models in $S_{test}$.

**Figure 8 genes-13-02344-f008:**
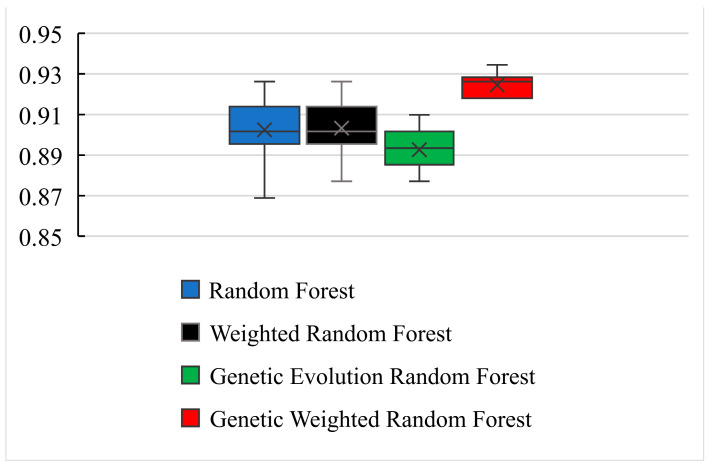
The boxplot of 4 models in 10 independent experiments.

**Figure 9 genes-13-02344-f009:**
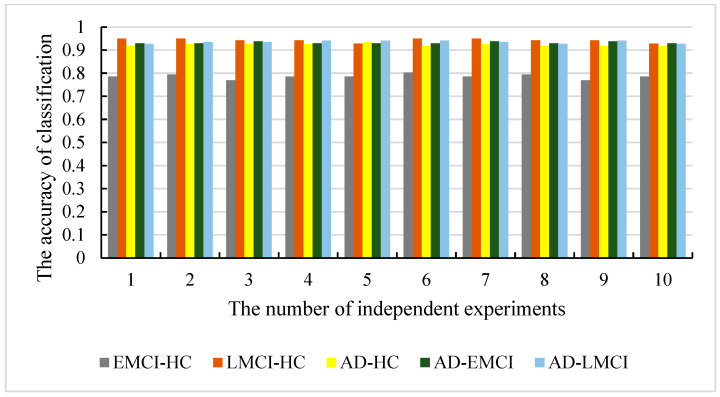
The classification accuracy of our models in five datasets.

**Figure 10 genes-13-02344-f010:**
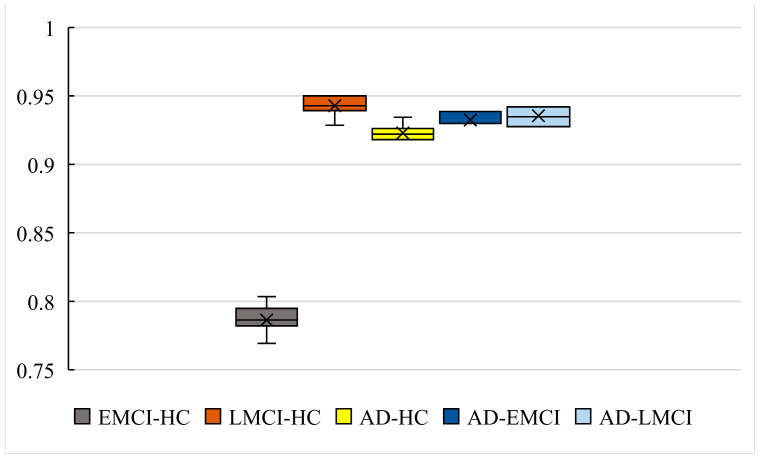
The boxplot of our models in five datasets.

**Figure 11 genes-13-02344-f011:**
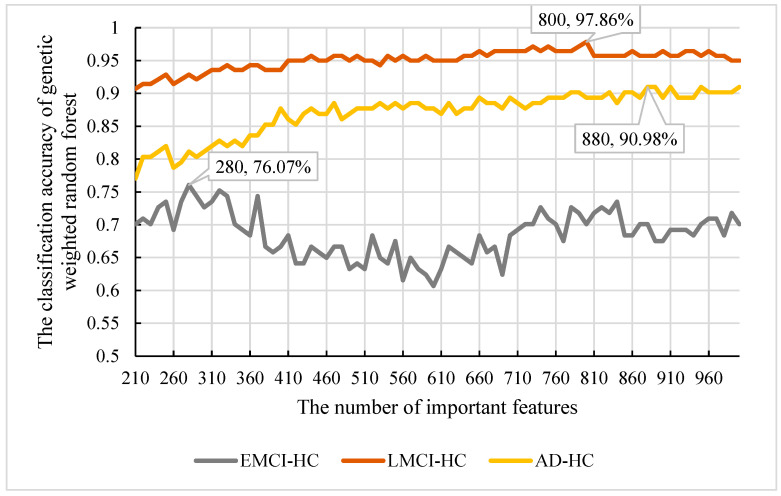
The classification accuracy and important features in $S_{AD-HC}$, $S_{LMCI-HC}$ and $S_{EMCI-HC}$.

**Figure 12 genes-13-02344-f012:**
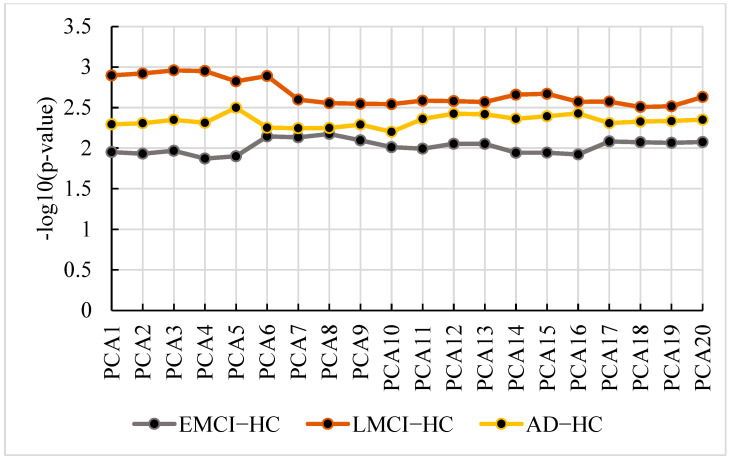
The *p*-value and PCA number in $S_{AD-HC}$, $S_{LMCI-HC}$ and $S_{EMCI-HC}$.

**Figure 13 genes-13-02344-f013:**
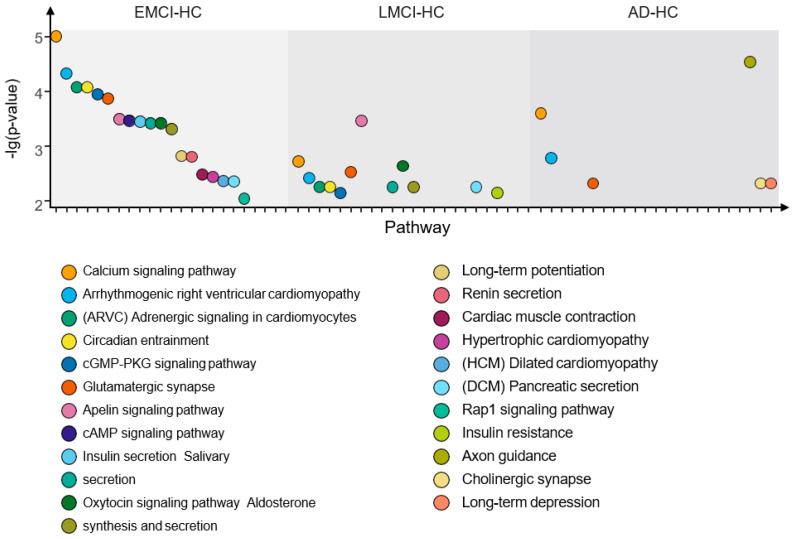
The pathways with corrected *p*-value < 0.01 of the EMCI−HC group, LMCI−HC group and AD−HC group.

**Figure 14 genes-13-02344-f014:**
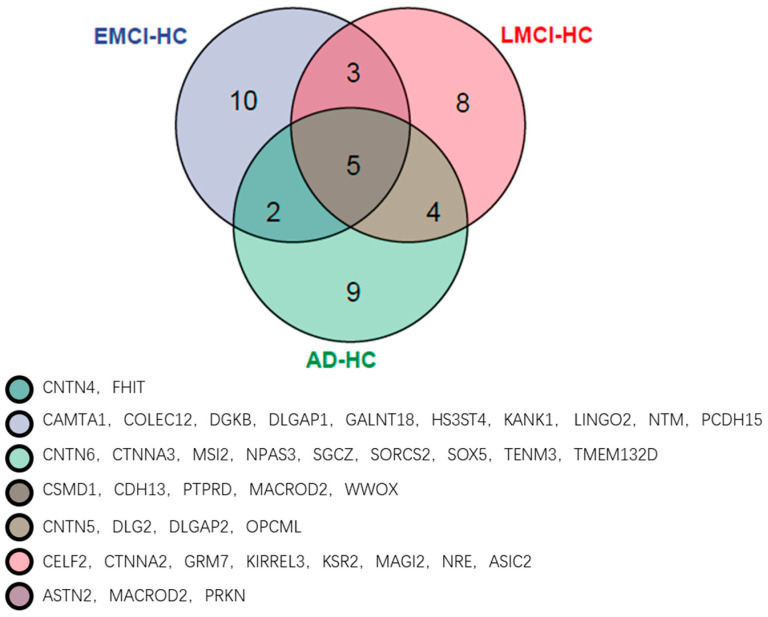
The top 20 genes of the EMCI−HC group, LMCI−HC group and AD−HC group.

**Table 1 genes-13-02344-t001:** Participant characteristics. HC = healthy control; EMCI = Early Mild Cognitive Impairment; LMCI = Late Mild cognitive Impairment; AD = Alzheimer’s disease; M/F = male/female; Edu = education; sd = standard deviation; *p* = *p*-value calculated by *t*-test.

Subjects	HC	EMCI	LMCI	AD	*p*
Number	310	271	390	296	-
Gender (M/F)	166/144	153/118	195/195	166/130	<0.001
Age (mean ± sd)	74.8 ± 5.4	71.3 ± 7.2	73.6 ± 7.6	75.2 ± 7.9	<0.001
Edu (mean ± sd)	16.3 ± 2.7	16.1 ± 2.6	15.8 ± 2.9	15.2 ± 3.0	<0.001

**Table 2 genes-13-02344-t002:** The Precision, Recall and F1 score of valid set and test set.

Methods	Valid Set			Test Set		
	Precision	Recall	F1 Score	Precision	Recall	F1 Score
Random Forest	0.76	0.75	0.76	0.89	0.88	0.88
Weighted Random Forest	0.76	0.75	0.76	0.89	0.88	0.88
Genetic Evolution Random Forest	0.73	0.72	0.72	0.85	0.84	0.84
Genetic Weighted Random Forest	0.82	0.81	0.81	0.89	0.88	0.88

**Table 3 genes-13-02344-t003:** The Precision, Recall and F1 score of our model in 10 independent experiments.

No.	EMCI-HC			LMCI-HC			AD-HC			AD-EMCI			AD-LMCI		
	Precision	Recall	F1 Score	Precision	Recall	F1 Score	Precision	Recall	F1 Score	Precision	Recall	F1 Score	Precision	Recall	F1 Score
1	0.76	0.73	0.73	0.92	0.91	0.91	0.86	0.86	0.86	0.87	0.87	0.87	0.88	0.88	0.88
2	0.79	0.75	0.75	0.95	0.95	0.95	0.8	0.8	0.8	0.83	0.83	0.83	0.86	0.87	0.86
3	0.76	0.74	0.74	0.93	0.91	0.92	0.89	0.88	0.88	0.85	0.85	0.85	0.87	0.88	0.87
4	0.76	0.73	0.73	0.93	0.92	0.92	0.86	0.86	0.86	0.86	0.86	0.86	0.86	0.86	0.86
5	0.74	0.72	0.72	0.93	0.92	0.92	0.85	0.85	0.85	0.84	0.84	0.84	0.85	0.86	0.85
6	0.72	0.7	0.7	0.93	0.91	0.92	0.85	0.85	0.85	0.88	0.88	0.88	0.88	0.89	0.88
7	0.77	0.75	0.75	0.95	0.94	0.94	0.86	0.86	0.86	0.85	0.85	0.85	0.87	0.87	0.87
8	0.81	0.76	0.76	0.93	0.91	0.92	0.83	0.83	0.83	0.86	0.86	0.86	0.87	0.87	0.87
9	0.77	0.75	0.75	0.92	0.91	0.91	0.85	0.85	0.85	0.86	0.86	0.86	0.87	0.88	0.87
10	0.79	0.72	0.72	0.93	0.92	0.92	0.84	0.84	0.84	0.85	0.85	0.85	0.88	0.89	0.88

## Data Availability

Data used for this study were obtained from ADNI studies via data sharing agreements that did not include permission to further share the data. Data from ADNI are available from the ADNI database (adni.loni.usc.edu) upon registration and compliance with the data usage agreement.

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
