# Peer review of "Detection of Association Features Based on Gene Eigenvalues and MRI Imaging Using Genetic Weighted Random Forest"

_genes, 2022, doi:10.3390/genes13122344_

Round 1
Reviewer 1 Report
The manuscript titled "Detection of Association Features Based on Gene Eigenvalues and MRI Imaging Using Genetic Weighted Random Forest" proposes a novel method to fuse features of gene eigenvalues and MRI images utilizing machine learning approaches (Genetic Weighted Random Forest). Regarding the results, five genes (CSMD1, CDH13, PTPRD, MACROD2, and WWOX) are in the intersection region of Alzheimer's disease (AD), late mild cognitive impairment (LMCI), and early mild cognitive impairment (EMCI). These findings have been supported using available studies in the literature properly. The topic is interesting, and the manuscript has been written well. However, there are some concerns as follows that should be addressed by the authors:
1- The missing values have been filled by zero from Equation 2. Does the 0 value affect the training process? Is it meaningless as a numerical value? Please justify this decision.
2- Please append the performance of independent experiments as boxplots to give better information about the model and data.
3- The accuracy metric is not adequate for classification problems to judge models. It is necessary to mention some other metrics, such as F1-score.
4- It is possible to merge various distributions and discriminate targets using Deep Learning as an alternative to the current study method. Please discuss it.
5- While a comparable performance is expected between validation and test, it seems that the AUC of S_valid (Figure 5) is significantly less than S_test (Figure 7). Please interpret these results.
Reviewer 2 Report
The authors have presented a novel method to integrate imaging and genomic data and perform feature detection. The authors do a good job of establishing the need for such a technique in the introduction section and then present their techniques on their data and present the results. The plots are well presented and explain the results of the paper clearly.
The clarity and coherence of the language in the methods section need to be significantly improved. This is much needed because the data used to reach this conclusion must be requested and is not publicly available. The authors should provide the accession ID for the data they have used in this study. Furthermore, the code files are also not provided, which makes it challenging to reproduce this work.
Since the authors present a novel algorithm for data fusion and feature detection, it would be highly beneficial to the community if they demonstrated their work on a publicly available dataset for the sake of reproducibility. This can be presented as an example within this paper or as an appendix to this paper.
Following are my requested changes/suggestions to the authors:
- Abstract: Line 15-16: Clarification is needed if LMCI, EMCI, HC, AD Progression are different categories/stages of AD.
- Introduction Line 35: Citations or examples of imaging/genetic data and explain why there is now a large amount of data available.
- Introduction Line 62-65: Please give examples of the appropriate models in place of the traditional machine learning models in the current data context. Explaining the limitations of traditional method. The authors mention about handling large number of features, they can explain why traditional approaches fail in such cases.
- Introduction Line 72: This is the first mention of the term "fusion features" in the article; please explain it briefly here as it is unclear to follow the idea conveyed here without reading the whole paper.
- Materials and methods Line 83-84: Please provide the data identifier/accession IDs that need to be requested.
- Feature Fusion Line 125: For consistency, the authors should rename the heading Fusion Features as they have in the rest of the text.
- Feature Fusion Line 128: Please explicitly explain how the size of M was computed.
- Feature Fusion Line 150: Typo correct "donate" to "denote."
- Genetic Weighted Random Forest Construction: Line 184: The authors should specify if they rank the decision trees based on validation or test accuracy.
- Genetic Weighted Random Forest Construction: Line 188: Please explain why the steps are done 50 times, if there is some type of convergence metric being evaluated ?
Reviewer 3 Report
The article proposes a method that uses data fusion and a genetic weighted random forest to mine features of Alzheimer's progression: Alzheimer's disease (AD), late mild cognitive impairment (LMCI), early mild cognitive impairment (EMCI) and healthy controls (HC). It uses as input genetic data (from GWAS) and MRI images. I suggest the authors to discuss the further utilization of this useful methodological approach, once it may be applied to other diseases, varying the input when needed.
Some background on the AD diagnosis would be welcome in the Introduction so that it would be clear to the readers the importance and applicability of this methodology. How the findings of this work could help in clinical practice? Could it be used to help identifying/differentiate features in other genetic diseases? These points should be mentioned.
I suggest the authors to review the English and the comma use (for exemple: L16, L26...).
L429-431 - Did the authors mean the phrase as it is?
Please, check the resolution of the Figures. Figures 1 and 4 present a very low definition. I suggest increasing and standardizing the font size within the figures.
Round 2
Reviewer 1 Report
The authors have addressed issues. I don`t have further comments.
Author Response
Thank you very much for your comments and suggestions.
Reviewer 2 Report
Thank you for making the suggested changes.
Author Response

(The authors gave the same response as above.)
